# Therapeutic Aspects and Molecular Targets of Autophagy to Control Pancreatic Cancer Management

**DOI:** 10.3390/biomedicines10061459

**Published:** 2022-06-20

**Authors:** Md. Ataur Rahman, Kazi Rejvee Ahmed, MD. Hasanur Rahman, Md. Anowar Khasru Parvez, In-Seon Lee, Bonglee Kim

**Affiliations:** 1Department of Pathology, College of Korean Medicine, Kyung Hee University, Hoegidong Dongdaemungu, Seoul 02447, Korea; kazirejveeahmed@gmail.com (K.R.A.); hasanurrahman.bge@gmail.com (M.H.R.); 2Korean Medicine-Based Drug Repositioning Cancer Research Center, College of Korean Medicine, Kyung Hee University, Seoul 02447, Korea; 3Global Biotechnology & Biomedical Research Network (GBBRN), Department of Biotechnology and Genetic Engineering, Faculty of Biological Sciences, Islamic University, Kushtia 7003, Bangladesh; 4Department of Microbiology, Jahangirnagar University, Savar, Dhaka 1342, Bangladesh; khasru73@juniv.edu; 5Acupuncture & Meridian Science Research Center, Kyung Hee University, Seoul 02447, Korea; inseon.lee@khu.ac.kr

**Keywords:** autophagy, pancreatic cancer, PC, autophagy, pancreatic ductal adenocarcinoma, PDAC, tumor-promoting, tumor-suppressive

## Abstract

Pancreatic cancer (PC) begins within the organ of the pancreas, which produces digestive enzymes, and is one of the formidable cancers for which appropriate treatment strategies are urgently needed. Autophagy occurs in the many chambers of PC tissue, including cancer cells, cancer-related fibroblasts, and immune cells, and can be fine-tuned by various promotive and suppressive signals. Consequently, the impacts of autophagy on pancreatic carcinogenesis and progression depend greatly on its stage and conditions. Autophagy inhibits the progress of preneoplastic damage during the initial phase. However, autophagy encourages tumor formation during the development phase. Several studies have reported that both a tumor-promoting and a tumor-suppressing function of autophagy in cancer that is likely cell-type dependent. However, autophagy is dispensable for pancreatic ductal adenocarcinoma (PDAC) growth, and clinical trials with autophagy inhibitors, either alone or in combination with other therapies, have had limited success. Autophagy’s dual mode of action makes it therapeutically challenging despite autophagy inhibitors providing increased longevity in medical studies, highlighting the need for a more rigorous review of current findings and more precise targeting strategies. Indeed, the role of autophagy in PC is complicated, and numerous factors must be considered when transitioning from bench to bedside. In this review, we summarize the evidence for the tumorigenic and protective role of autophagy in PC tumorigenesis and describe recent advances in the understanding of how autophagy may be regulated and controlled in PDAC.

## 1. Introduction

Pancreatic cancer (PC) is amongst the most deadly cancers because of late detection, high metastatic potential, and invasive regional advancement [1]. Despite medicinal and diagnostic improvements in all other forms of cancer, clinical management of PC remains restricted, posing a serious health risk to patients [2]. Indeed, the life expectancy for PC patients remains at only 6–9 months, with a 5-year survival rate of only 9% [3]. Pancreatic ductal adenocarcinoma (PDAC) is the dominant type of PC and is associated with many genetic alterations [4]. Common genetic mutations found in PC that likely enhance its tumorigenesis are present genes encoding Kirsten rat sarcoma virus (*KRAS*), tumor protein P93 (*TP53*), cyclin-dependent kinase inhibitor 2A (*CDKN2A*), and SMAD family member 4 (*SMAD4*) [5]. These mutations influence tumorigenesis in both cell-autonomic and non-cell-autonomic forms, adversely affecting protein synthesis, cell division, cell metabolism, and cell propagation, which collectively lead to cellular instability [6].

Autophagy plays a crucial role in preserving intracellular homeostasis [7,8]. It performs a wide range of functions, including the conservation and protection of cellular activities, the release of basic components, and the assistance of metabolism-related activities, where intracellular compounds are engulfed by double-membraned autophagosomes and transported to lysosomes for degradation [9]. Recent technological advancements have enabled the discovery of additional autophagy controller proteins, substances, and receptors in yeast and mammalian cells, advancing understanding of the process [10,11]. Macroautophagy, chaperone-mediated autophagy (CMA), and microautophagy are the three main types of autophagy [12]. Autophagy can selectively and precisely target multiple cell compartments almost simultaneously, including mitochondria, endoplasmic reticulum, peroxisomes, nucleus, lysosomes, and even other cellular organelles such as fat droplets and accumulates, to maintain cell stability under both normal and abnormal conditions [13,14]. Many studies have individually examined macroautophagy, targeted autophagy, and autophagy regulators to broaden our knowledge of how autophagy functions in PC [15,16]. This review focuses on the role of autophagy in PC tumor formation, aggressiveness, and medicinal weaknesses.

Under normal physiological conditions, in the initial stage of PC, tumor cells show high autophagy levels, indicated by increased numbers of lipidated microtubule-associated protein light chain 3 (LC3-II) and autophagosomes [17]. Autophagy inhibition through RNA interference or tiny particular molecule blockers has been shown to reduce damage in PDAC cells and tumor size in PDAC xenografts [18]. Nevertheless, autophagy appears important for PDAC tumor development, and drug trials using autophagy inhibitors, either alone or in combination with other treatments, have had limited success [19]. In some tumors, an increase in reactive oxygen species (ROS) levels induces cell death in PDAC cell lines, accompanied by a decrease in autophagy levels [20]. This observation highlights the intricacies of autophagy control and its consequences in PDAC cells. Moreover, a large body of evidence exists indicating that autophagy is required for tumor metabolism and survival. In this review, we describe the current understanding of the functional role and regulation of autophagy in PDAC.

## 2. Methods

A literature-based search of publication databases was performed to identify relevant original English-language articles relating to PC therapeutic targets, prevention, and molecular mechanisms. The databases included PubMed, Scopus, Web of Science, and Google Scholar. The searches were performed using various keywords, including pancreatic cancer, autophagy, apoptosis, natural compounds, pancreatic cancer therapy, phytochemical, drug delivery system, targeted signaling pathway, and perspectives role of pancreatic cancer treatment. All figures were created using Adobe Illustrator software (San Jose, CA, USA).

## 3. Molecular Pathway of the Autophagy Process

Autophagy can be triggered in response to faulty cytoplasmic components such as protein clusters, and mitochondrial damage [21,22]. However, its primary function is maintaining intracellular homeostasis under stress and injury conditions [23]. UNC51-like kinase 1 (ULK1) and its initial protein complex are regulated primarily by metabolic sensor processes, including the mammalian target of rapamycin complex 1 (mTORC1) and AMP-activated kinase (AMPK), which negatively and positively regulate autophagy, respectively [24]. The phosphatidylinositol 3-kinase (PI3K) type III nucleation complex containing beclin 1 (BECN1) is then activated to stimulate the nucleation of pre-phagophores with the assistance of autophagy-related 9 (ATG9)-bound vacuoles [25]. Lipidation of LC3-II (autophagy-related 8 [ATG8]) protein has a binding site that attaches to the LC3-interacting motifs (LIRs) on autophagosome carriers, attaching it to the pre-phagosome [26]. Their binding with phosphatidylethanolamine (PE) is performed with the assistance of autophagy-related 3 (ATG3), 4B (ATG4B), and 7 (ATG7), an autophagy-related 12 (ATG12)-conjugated complex, and the phosphatidylinositol 3,4,5-triphosphate (PIP3) binding complex [27]. Initially, LC3 is cleaved by ATG4B to uncover a glycine, creating LC3-I, which is then transferred to the E1-like enzyme ATG7 complexed with ATG12, followed by the E2-like enzyme ATG3, attaching it to the main PE group, creating LC3-II [28,29]. ATG7 is activated by ATG12 and forms intermediates with ATG7 and E2-like enzyme ATG10 before conjugating with ATG5 [30]. ATG12/ATG5 complexes function similar to E3-like enzymes to promote the formation of ATG8-PE complexes [31]. Autophagy-related 16-like 1 (ATG16L) is required for this innate process, but it does ensure that lipidation occurs on the appropriate cell wall. These actions cause the phagophore to enlarge and close, controlling the formation of the autophagosome [32]. The autophagosome’s two layers merge with lysosomes just before closing, permitting degradation of the inner membrane and intra-autophagosome transfer to begin [33].

A detailed molecular mechanism of autophagy is presented in Figure 1. Various drugs can manipulate autophagy, creating significant pharmacological interest in PDAC regulation. PC cells undergo a metabolic switch from oxidative phosphorylation to glycolysis, causing them to synthesize adenosine triphosphate (ATP) from glucose via glycolysis without oxygen [34]. It has been reported that the relationship between energy metabolism and autophagy creates a relationship between autophagic activity and cellular ATP levels [35,36]. However, inhibition of ATP production via carbon source deprivation or ATP synthesis perturbation suppresses autophagic activity [37]. In addition, glucose starvation does not activate autophagy, and AMPK was dispensable in amino acid starvation-mediated autophagy, indicating the absence of autophagy signaling machinery in low-energy states [38]. Numerous pathways, such as the ADP ribosylation factor like GTPase 6 interacting protein 5 (ARL6IP5, also known as JWA), AMPK, forkhead box O3 (FOXO3a), and protein tyrosine kinase 2 (PTK2, also known as FAK) axis and the KRAS, TP53, and TP53 induced glycolysis regulatory phosphatase (TIGAR) axis, are all implicated in the reprogramming of glycolysis that occurs in PC cells [34]. Therefore, these signaling pathways are potential PC therapeutic targets and play a crucial role in glycolysis.

## 4. Molecular Mechanism and Formation of Pancreatic Adenocarcinoma

PDAC has the tenth highest incidence of all cancer types but is the fourth leading cause of cancer-related deaths, with a comparative one-year life expectancy of only 24% [39]. PDAC is aggressive during its early stages, with local recurrence and remote metastases [40]. The growth of PDAC involves a steady accumulation of gene mutations [41]. The first genetic mutation detected was in *KRAS* and is found in approximately 90–100% of PDACs [42]. The mutant KRAS protein creates signals for expansion, differentiation, and viability [43]. Consequently, *KRAS* mutations are believed to represent the very first genetic changes in PDAC. Furthermore, homozygous loss of the 9q21 locus is present in almost 40% of PDAC tumors [39]. The 9q21 locus contains the cyclin-dependent kinase inhibitor 2A(*CDKN2A*) gene encoding the p16^INK4a^ and p19^ARF^ tumor inhibitory proteins, created through alternate first exons and reading frames, which play a vital role in PDAC development [39]. In the late stages of PDAC formation, *p53* oncogene mutations and SMAD4 degradation are common [13,44]. PDAC begins to develop anatomically from a predecessor wound called “pancreas intraepithelial neoplasias” (PanINs) [45]. PanINs exhibit glands-like structures with duct-like features and varying levels of abnormality in cells and differentiation and are categorized from level I, which has columnar cylindric mucinous epithelium, to grades levels II and III, which have nuclear abnormalities. However, higher-level PanINs convert into PDAC when damaged areas stretch beyond the basement membrane [46]. Eventually, *p53* mutations and SMAD4 degradation are prevalent in the final stages of PDAC [47]. Several other signaling pathways may be involved in modulating PDAC formation in PC (Figure 2). KRAS plays a central role in modulating cellular differentiation, proliferation, and apoptosis in PDAC.

## 5. Interactions for Both Autophagy and PDAC

In vivo and in vitro studies have shown that autophagy tends to impact the growth and preservation of PC cells [48]. Nevertheless, even within the same study, results may show autophagy to have both a pro- and anti-survival function [49]. However, both PDAC primary tumors and cell lines show high basal autophagy, the functional significance of which remains unknown [50]. Genetic and pharmacologic autophagy impairment has been used to inhibit PDAC cell line growth in vitro, possibly due to higher levels of ROS and DNA damage and reduced mitochondrial activity [50,51].

Across PC xenograft and mice models, autophagy inhibition via small hairpin RNA (shRNA) or small molecule inhibitors targeting ATG5 causes tumor regression and longer survival because ATG5 and ATG7 proteins are required for autophagosome formation [8,52], and *ATG5* or *ATG7* gene deletions reduce autophagy effects [53]. A mouse model with carcinogenic *KRAS* expression in pancreatic cells in the presence of two, one, or no *Atg5* gene copies found that reduced Atg5 protein levels enhance tumor formation, but the absence of Atg5 prevents tumorigenesis [54]. Therefore, when primary PC cell lines lacking Atg5 were injected into a mouse, they observed improved intrusive and metastatic capability. Human PDAC samples have also been studied, showing that lower ATG5 levels were associated with tumor metastasis and lower survival [55]. A single copy *Atg5* deletion with the autophagy suppressor chloroquine (CQ) caused increased aggressiveness and metastatic expansion, with significant diagnostic consequences since CQ may increase the risk of creating resistant cancer cells, enhancing its aggressiveness [56]. This is significant because whether autophagy has a tumor-inhibiting or -enhancing role is likely event-dependent [57].

Genetically engineered PDAC mice missing the important autophagy genes *Atg5* or *Atg7* form low-grade premalignant pancreatic lesions, indicating that autophagy continues to play a defensive function in the initial stages of tumorigenesis [15]. However, autophagy speeds up tumor initiation rather than preventing tumor formation in a mouse model with carcinogenic KRAS and homozygous deletion of *P53*, with tumor proliferation enhanced by greater glucose ingestion [58]. Furthermore, the deletion of Atg5 or Atg7 in the pancreas in the presence of a triggering *KRAS* mutation prevented premalignant damage from progressing to cancer [57]. One limitation of these studies has been their use of a homozygous *P53* deletion framework that may not be fully representative of human tumors in which only one copy of *P53* is inactivated [59].

The appearance of therapy-resistant cancer stem cells with self-renewal and mobility is an essential characteristic of PDAC. A study comparing PDAC cells with increased levels of stem cell signs showed that cells with higher levels had elevated levels of autophagy [56]. Apoptotic cell death of PDAC stem cells and a decline in migratory effect and tumorigenicity all occurred due to autophagy inactivation. Hypoxia has been used to encourage the infective and stem-cell-like properties of PDAC cell lines and to increase autophagy indicators such as BECN1 and LC3-II [60]. Therefore, these observations indicate that autophagy may enhance cancer-stem-cell metastatic potential under a hypoxic PDAC cellular environment. A subsequent study highlighted the link between autophagy and PDAC stem cells by assessing stem cell indicators such as aldehyde dehydrogenase (ALDH1) in PC tissues from sick people using gene expression microarrays [61]. Individuals with elevated co-expression of autophagy indicators LC3-II and ALDH1 had inadequate sustainability, indicating that survival is advancement free [56]. Another cell line-based study confirmed these findings, showing that, compared to sphere-forming stem cells, PDAC cells have elevated levels of the autophagy indicator LC3-II. Therefore, the function of autophagy in PDAC survival does not depend on the tumor cells but on their microenvironment. 

Amino acids that are not essential for PDAC metabolism are secreted by stroma-related pancreatic asteroid cells [62]. Cancer cells stimulate autophagy in star-like shaped cells, leading stellate stroma cells to release alanine, which can act as a substitute carbon source for cancer cells. This switch is essential for PDAC tumors, providing a microenvironment that is insufficient in glucose and nutrients from serum [63]. Moreover, evidence indicates that autophagy may play a role in preventing tumorigenesis inside the pancreas. For example, autophagy has been shown to help stop endoplasmic reticulum (ER) tension in pancreatic acinar cells [64], which are exocrine pancreatic cells that produce and contain a substantial number of digestion enzymes. As acinar cells induce cells to secrete pancreatic enzymes, those with a large ER channel and burden have also been associated with pancreatitis, a potential risk factor for PDAC development. Mice missing *ATG7*, which is required for autophagy, have decreased autophagic flux and increased ER strain, acinar cell degradation, and pancreatic inflammatory responses [65].

Autophagy in PC cells is triggered via multiple signals as a response to external stressors such as cytotoxic drugs, hypoxia, radiation, and nutrient deprivation (Figure 3), increasing metabolism in PDAC cells by providing energy and building blocks to help sustain proliferation. Therefore, autophagy has an important function in the maturation and maintenance of T-cell antigen presentation through macrophages in addition to activation of other antigen-presenting cells.

### 5.1. Positive Role of Autophagy in PDAC Development

In general, 95% of PDAC carry KRAS mutations [66], and a recent anti-KRAS treatment study is pursuing other options. To determine whether KRAS deficiency contributes to the elevated basal autophagy levels observed in KRAS mutant PDAC, it was repressed in various mouse and human PDAC cell lines via small interfering RNA (siRNA) and small-molecule inhibitors targeting the KRAS effector extracellular signal-regulated kinase 1/2 (ERK) [67]. Surprisingly, inhibiting KRAS and ERK led to increased autophagic flux but reduced glycolytic and mitochondrial capability. This finding suggests that inhibiting ERK increases the reliance of PDAC on autophagy and that the pharmacologic blocking of both ERK and autophagy may represent an efficient PDAC therapy [67]. This possibility is supported by experiments in which the autophagy inhibitor CQ was used synergistically with ERK blockers [67].

In addition, studies using siRNA to explore dependency trends in KRAS mutated and wildtype PC cell lines showed that targeting the B-Raf (BRAF) and Raf-1 (CRAF) serine/threonine kinases in combination with the autophagy E1 ligase ATG7 effectively eradicated KRAS mutant cell types with minimal toxic effects on normal cells [68]. Therefore, significant evidence from various sources indicates that the mitogen-activated protein kinase (MAPK) and autophagy processes cooperate to preserve KRAS mutant tumor cells. One possible mechanism through which oncogenic KRAS controls autophagy in PDAC is via the activation of vacuole-related type membrane protein 1 (VMP1), which is required for autophagosome creation [69]. RNA inhibition (RNAi) studies have shown that VMP1 is required for KRAS to trigger and preserve autophagy, indicating that this process is mediated by GLI family zinc finger 3 (GLI3) and transcription controlled by the hedgehog pathway, stimulating the transcription of VMP1 [69].

Similarly, an electron microscopy ultrastructural analysis showed that cells lacking VMP1 are capable of generating small immature autophagosome-like structures but are unable to extend or progress inside autophagosomes [70]. Osteopontin (OPN) is a glycoprotein that performs roles in cancer development and autophagy initiation in smooth muscle cells via the integrin/CD44 and p38 MAPK routes [71]. In studies of PDAC stem cells, it was found that OPN encourages LC3-II protein expression, increasing the LC3-II/LC3-I ratio and autophagic flux, and encouraging stem cell signs such as ALDH1 and CD44. Prevention of nuclear factor kappa B (NF-kB) activation might also inhibit OPN-induced autophagy independent of other OPN signaling pathways [71].

While gemcitabine is one of the standard chemotherapeutic treatments used for PDAC, resistance is a common occurrence [72]. Inhibiting autophagy reduces the number of pancreatic stem cells and their capacity to establish spheres and enhances the sensitivity of PDAC cells to gemcitabine. Long non-coding RNAs (lncRNAs) may be involved in the regulation of cell death in cancer cells [73]. For example, the small nucleolar RNA host gene 14 (SNHG14) plays a role in cancer development in various cancer types and is highly expressed in PDAC compared to normal tissue [74]. SNHG14 has been shown to interact with the adverse autophagy controller microRNA miR-101, potentially decreasing miR-101 stages and promoting autophagy, increasing the sensitivity of PDAC cells to gemcitabine [74,75].

### 5.2. Adverse Role of Autophagy in PDAC Development

Lysosomal-associated membrane proteins 1 (LAMP1) and 2 (LAMP2) are controllers of autophagosome development and the primary components of the lysosomal membrane [76]. In addition, studies have shown that alterations in PDAC can induce lysosome biogenesis transcription-associated activation [77]. There is a growing body of evidence indicating that ubiquitin-type protein 4A (UBL4A) interacts with LAMP1. UBL4A is a tumor suppressor that mediates DNA damage response and is a protein-folding chaperone [78]. The connection between UBL4A and LAMP1 has been shown to affect lysosome operation, with autolysosome accumulation and lysosomal dysfunction in neurons with increased UBL4A levels [79]. However, an assessment of UBL4A expression in 69 PDAC patients showed that patients with increased UBL4A expression in PDAC tissues had increased survival [56], potentially due to reduced autophagy.

Optineurin (OPTN) is a cell wall trafficking protein that functions as a luggage binding site in specific autophagy, transporting polyubiquitinated cargo to the autophagosome through its LC3-related binding site [80]. OPTN and other autophagy proteins, including LC3 and GABA type A receptor-associated protein-like 2 (GABARAPL2), are abundantly expressed in PC based on the Human Protein Atlas data [81]. A panel of PDAC cell types was used to investigate OPTN, showing that siRNA knockdown of OPTN did not significantly affect cell survival [80]. Interestingly, these results also indicated that silencing OPTN adversely affected macroautophagy stages, indicating higher chaperone-related autophagy.

Precursor neural growth factor (proNGF) is a controller of neuronal regeneration and advancement that is abundantly expressed in various cancerous tumor cells [82]. In PDAC cell lines, siRNA-mediated knockdown of proNGF decreases levels of ATG5 and BECN1 and increases levels of p62, an autophagy-degraded cargo protein, indicating that reduced proNGF may repress autophagy. In addition, it has been shown that autophagy is required for PDAC cell growth, movement, and pervasiveness.

## 6. Autophagy-Based Treatment Strategy for PDAC

PC remains one of the most deadly and challenging diseases to treat and cure. However, advances in understanding the biological basis of autophagy and its roles in cancer, the microenvironment, and the macroenvironment have provided patients with innovative treatments and medicines. Before developing selective autophagy blockers, scientists attempted to influence autophagic flux via upstream processes. Ozpolat et al. discovered that protein kinase C-delta (PKCδ) prevents autophagy via transglutaminase 2 (TG2) activation, a marker of metastasis and poor patient prognosis in PC. Inhibiting the PKCδ/TG2 axis induced autophagy, identifying PKCδ/TG2 as a potential new therapeutic avenue [83]. In addition, the PKC delta blocker Rottlerin induced intrinsic and extrinsic apoptosis in PC. However, this effect was based on eukaryotic elongation factor-2 kinase (eEF2K), not PKCδ. Nevertheless, Rottlerin inhibited eEF2K expression and promoted its ubiquitin-mediated proteasomal degradation [84].

PC also shows genomic disturbances and epigenetic changes, such as DNA methylation and histone alteration. Notably, histone deacetylase (HDAC) blockers affect epigenetic and autophagic mechanisms. The promising therapeutic agent and HDAC blocker suberoylanilide hydroxamic acid (SAHA) stimulated autophagy in cancer, inhibiting the Akt/mTOR pathway and triggering the ER-stress response [85,86]. In addition, SAHA enhanced the sensitivity of PC cells to gemcitabine. Cell death induced by the autophagy-causing agent triptolide resulted in the eradication of metastatic PC cells via inhibition of the Akt, mTOR, and ribosomal protein S6 kinase B1 (RPS6KB1, also known as p70^S6K^) axis and promotion of the ERK1/2 pathways [87].

Repurposing medically proven medications is a simple and realistic therapeutic strategy in PC. For instance, omeprazole inhibited the proliferation of PC cells at non-cytotoxic levels, restored the biphasic impact of 5-fluorouracil (5-FU), and regulated the lysosomal delivery process, impairing autophagy and causing programmed cell death [88]. Consistent with these findings, autophagy inhibition reduced PC cell proliferation following 5-FU and gemcitabine [89]. Furthermore, recent discoveries on the function of autophagy in PC models prompted experiments using hydroxychloroquine (HCQ), a well-known therapeutic drug, as a neoadjuvant treatment in PC patients (Figure 4). Regular application of HCQ to patients with a history of metastatic PC resulted in unstable autophagy inhibition and negligible therapeutic benefits in PDAC patients [90]. In addition, combining HCQ with the well-established PC treatments gemcitabine and nab-paclitaxel (GA) in mature PC patients reduced their 12-month life expectancy rate and mean overall survival compared with patients not treated with HCQ. While the mixture of HCQ and GA reduced some of its adverse effects, such as neutropenia and anemia, it exacerbated others, such as weakness, nausea, peripheral neuropathy, facial changes, and neuropsychiatric symptoms [91]. Following care, the genomic review of patients showed no significant association with *p53* mutation status. Therefore, the results of pioneering studies that established a relationship between *p53* mutation and autophagy function in PC have not yet been successfully translated into medical practice.

Subsequent studies that have used a mixture of ERK and autophagy blockers showed impressive results. Autophagy was induced by inhibiting KRAS messaging or its downstream effectors, shielding PC cells from cytotoxic activity [67,92]. In cancerous cells, trametinib inhibited MAPK/ERK kinases 1 and 2 (MEK1/2) in a time- and dose-dependent manner, activating the liver kinase B1 (LKB1), AMPK, and ULK1 axis and autophagy. Stable KRAS knockdown also enhanced autophagic flux, as shown by doxycycline elimination from murine PDAC cells harboring doxycycline-inducible KRASG12D [93]. Similarly, ERK inhibition increased the reliance of PDAC cells on autophagy, most likely through effects on lysosomal acidity, glucose breakdown, and mitochondrial biogenesis. 

Preclinical tumor models immune to treatment with CQ or trametinib alone were highly sensitive to their mixed therapy. The translation of these results with one patient confirmed their combined trametinib and HCQ antitumor activity, decreasing tumor burden and marker CA19-9 and resolving metastatic liver lesions [93]. However, PDAC may recur, and patients should be monitored for an extended period of time. In addition, evidence of the therapeutic efficacy of this combinatorial treatment should be obtained in a larger patient cohort (Figure 4). Therefore, the use of autophagy-targeting drugs in PC is still dependent upon effective and appropriate findings from clinical trials.

Recently, the European Medicines Agency (EMA) recommended that Abraxane be approved for treating patients with metastatic PC [94]. Abraxane with gemcitabine is recommended for treating adults with metastatic PC. Abraxane contains the active material paclitaxel, a type of taxane, as an anticancer drug that kills cells by blocking cell division, which is attached to albumin to form an injectable solution [95]. However, the mTOR inhibitor rapamycin and its derivatives, rapalogues, are safe and effective in treating PDAC in clinical experience [96]. In addition, numerous autophagy inhibitors, such as the Lys05 family, ROC-30596, and GNS56197, are in various stages of clinical development for PDAC treatment [97]. Moreover, the US Food and Drug Administration (FDA) has approved several single drugs for wide use in PC treatment, such as erlotinib hydrochloride, Everolimus, 5-FU, gemcitabine hydrochloride, Gemzar (gemcitabine hydrochloride), Infugem (gemcitabine hydrochloride), irinotecan hydrochloride liposome, Lynparza (Olaparib), Mitomycin, and Olaparib [98]. Additionally, drug combinations used in PC are Folfirinox, gemcitabine-cisplatin, and gemcitabine-oxaliplatin [99]. Finally, the drugs approved for gastroenteropancreatic neuroendocrine cancers are lanreotide acetate, Lutathera (lutetium lu 177-dotatate), lutetium lu 177-dotatate, and somatuline depot [100].

## 7. Pharmacological Modulation of Autophagy in PDAC Regulation

The autophagy and tumor progression relationship are complex, despite autophagy having been shown to inhibit cancer initiation in numerous models. Therefore, developing a model explicitly designed to test the effects of autophagy inhibition in adult mice was needed. The adverse effects of autophagy inhibition, particularly neurodegeneration, reported with models in which key autophagy genes are deleted raise concerns about the safety of this approach, including increased infectivity, glucose imbalance, heart problems, and muscle, liver, and pancreas tissue damage [101]. However, while modest activation of autophagy has also been shown to boost life expectancy in some models [102], it has also been associated with increased toxicity, particularly cardiac toxicity.

In general, non-pharmacological autophagy-inducing behaviors such as exercise and caloric limitation via fasting and dieting were shown to benefit overall health [103]. Compared to genome editing, inactivation is rarely precise, fulfilled, and conclusive. Therefore, the impacts of genetic deletion of key autophagy genes are unlikely also to be reproduced precisely by pharmacological inhibition [104,105]. The precision of autophagy inhibition via pharmacological agents remains an open question complicated by the multiple autophagy-independent features of autophagy-related genes, even though some autophagy-independent mechanisms rely on scaffolding behaviors instead of catalytic activities [106] that may be agreeable to pharmacological modulation. Further studies are required to provide a high-level overview of this area supported by other in-depth studies on pharmacological autophagy modulation [107,108].

### 7.1. Autophagy Inhibition in PDAC Regulation

CQ and HCQ have both been utilized over a long period to inhibit autophagy in laboratory and clinical experiments despite each impairing lysosomal characteristics [109], causing endosomal depletion and vesicular trafficking. Similarly, numerous experiments have shown that the antitumor action of CQ and HCQ may be dissociated from their autophagy-inhibitory properties [110]. These clinical-stage autophagy inhibitors and others, including the Lys05 family, ROC-30596, and GNS561, target lysosomes. Clearly, HCQ and other lysosomotropic drugs suppress other cell processes associated with the lysosome, such as micropinocytosis, which may inhibit tumor growth, cause lysosomal permeation, and have a potent antitumor effect independent of autophagy [111,112,113]. Despite CQ and HCQ having been approved for a long time and being widely available [114], only a few trials have been conducted to investigate their use in PC patients.

Since the 1990s, most of the proposed autophagy pharmacological targets have been kinases, which are easily targetable with small molecules with substantial contribution and efficacy [115]. However, few compounds have translated into therapeutic use due to the uncertainty and debate concerning the effect of autophagy on cancer [116]. The ULK1 kinase complex, which includes the serine/threonine kinase ULK1 (ATG1), ATG7, and RB1 inducible coiled-coil 1 (RB1CC1, also known as FIP200), is activated in response to nutrient deficiency and is required for the induction of deprivation autophagy [117]. Numerous ULK1-related inhibitors have been developed (e.g., SBI-0206965), but none have advanced to clinical trials [118]. Similar to ATG7 inhibitors, none have been approved for therapeutic use.

PI3K catalytic subunit type 3 (PIK3C3, also known as VPS34) is a lipid kinase required for autophagy induction [119] and a participant in the PI3K pathway against which many broad-spectrum (3-methyladenine, wortmannin, and LY294002) and isoform-specific (alpelisib and idelalisib) inhibitors have been developed [120]. Over the last decade, new VPS34 inhibitors such as SAR-405 have been reported [121]. However, the development of several has been stopped due to the controversy surrounding the effect of autophagy on cancer. Nevertheless, some companies intend to improve VPS34 inhibitors for use in combination with a specific target and immune treatments. ATG4B is a cysteine protease that promotes autophagosome creation by converting LC3/ATG8 paralogues into their active state (LC3-I) by exposing the PE conjugation area [122] and is responsible for the deconjugation (delipidation) process that enables LC3 reuse [123]. While a dominant-negative form of ATG4B (C74A) sequesters LC3, knocking out *ATG4B* results in a temporary inhibition of autophagy due to rescue by other ATG4 isoforms. Consequently, the relative efficacy of isoform-specific versus pan-ATG4 inhibition in targeting autophagy remains debatable and will require further research. Similar to other autophagy objectives, while many ATG4B inhibitors have been reported over the last decade, none are currently in PDAC clinical trials.

### 7.2. Autophagy Activation in PDAC Regulation

While the majority of current attempts are directed at impairing autophagy in PC, the question of whether to restrict or stimulate autophagy remains. Autophagy activation using rapamycin-analog-mediated mTOR inhibition has highlighted an approach for investigating the use of PDAC models in clinical trials via inhibition of the PI3K/AKT signaling pathway [33,124]. However, while these results have not been interpreted as single agents, they show clinically important behavior in humans. One possibility is that inhibiting mTORC1 initiates a feedback loop resulting in AKT phosphorylation [125], with new recruits or interactions that circumvent this feedback loop potentially involved in PDAC [126]. However, mTORC1 has numerous phosphorylation targets in addition to the ULK1 complex, limiting the conclusions that can be drawn regarding autophagy stimulation by rapamycin analogs and other mTOR blockers.

## 8. Limitations and Prospective Regulation of PDAC via Autophagy

Various studies have shown that autophagy plays a pro-tumor survival role in metastatic tumors in PDAC, mainly through the changes in energy production and intermediates like alanine [127]. While PC tumor cells and murine xenografts show that inhibiting autophagy reduces proliferation and tumor size and prolongs survival, autophagy agonist treatments have not shown improvements over standard-of-care treatment. Autophagy inhibition alone is inadequate to inhibit PC proliferation, and combined treatments will almost definitely be needed to induce therapeutic reactions [108]. Considering current evidence on how ATG5 may affect pharmacological autophagy suppression results, it will be important to conduct correlation analyses between patient reactions to pharmaceutic autophagy inhibition and levels of ATG5 and other autophagy-related proteins in the search for diagnostic and medical approaches in PC [128,129].

Another potential approach is to observe the ratio at which autophagy is inhibited and determine whether it varies between patient cohorts. All approaches that can be taken here use pharmacodynamic indicators to determine the effectiveness of cell-death inhibition. However, one potential marker is LC3-II, which is monitored closely in human external lymphocytes after HCQ therapy [90]. Autophagy enables tumor cells to adapt to various stresses and strains, such as preserving energy stability and nutritional reserves under uncontrolled proliferation and detrimental microenvironmental factors such as low oxygen levels (hypoxia), pH, and nutrient availability [130]. Therefore, it may confer a degree of reliance of tumor cells on constant autophagy. While inhibiting autophagy was initially believed to be the way forward due to autophagic flux often being elevated in PDAC, the consequences of autophagy inhibition on the immune system and feedback mechanisms cast doubt on this, opening the way for novel therapeutic approaches. The impact of autophagy inhibition on the anticancer immune response must be evaluated in humans since mouse models have previously shown limited predictive value due to significant differences in how the murine and human immune systems operate and inherent limitations of inhibitors and genetically constructed designs.

## 9. Conclusions

Autophagy has long been considered a potential therapeutic target for PDAC. Nevertheless, a repertoire of safe and widely available compounds targeting several key autophagy elements is still required to advance our understanding of the effects of pharmacological autophagy inhibition and to enable its translation to human clinical use. Autophagy is believed to be one method through which tumor cells attain a high metabolic rate in nutrient-deficient surroundings. Moreover, it appears that tumor cells can also utilize autophagy from neighboring cells to preserve their nutrient supply. Additionally, several findings indicate that autophagy levels are regulated by a nutrient and energy balance, even in tumor cells. Therefore, time-dependent autophagy inhibition may be a potential strategy for disrupting tumor homeostasis [115].

The fact that autophagy has been shown to both promote and inhibit PDAC tumors does not preclude its therapeutic inhibition. The overwhelming balance of evidence suggests that autophagy should be inhibited in PDAC, and outcomes from combined inhibition of the ERK/MAPK pathway and autophagy have led to several clinical studies evaluating various ERK/MAPK-pathway inhibitors in combination with HCQ [67]. Autophagy studies in humans continue to be impeded by restrictions on instruments currently available for use. Consequently, many PCs need autophagy to be active all the time and targeting this process may represent a suitable treatment option. However, because PDACs have many feedback loops, systems that talk to one another, and parallel energy supply systems, it might be difficult to impede their energy metabolism by stopping autophagy alone.

Early clinical trials have shown that autophagy inhibition may not be sufficient as a standalone treatment. However, ongoing clinical trials are using a combination of an autophagy inhibitor with chemotherapy. Therefore, it would be beneficial to identify new, more effective approaches for inhibiting autophagy and biomarkers for monitoring autophagy inhibition in PDAC. The substantial body of evidence suggests that the constitutive activation of autophagy in PDAC tumors is a key contributing factor to the aggressive nature of PC and treatment resistance in patients. If we are to exploit autophagy activity as a weakness, it will be important to monitor autophagic dependency at the time of diagnosis for each tumor. This information will allow for a more accurate determination of the optimal therapeutic strategy combining autophagy inhibition with conventional chemotherapy to treat PDAC. Clinical trials investigating the biological effects of currently available autophagy-inhibiting compounds will advance our understanding of how they work in patients and enable improvements to preclinical PDAC models that will make them more accurate predictors. Ultimately, the effects of autophagy inhibition on PDAC will have to be evaluated in humans.

## Figures and Tables

**Figure 1 biomedicines-10-01459-f001:**
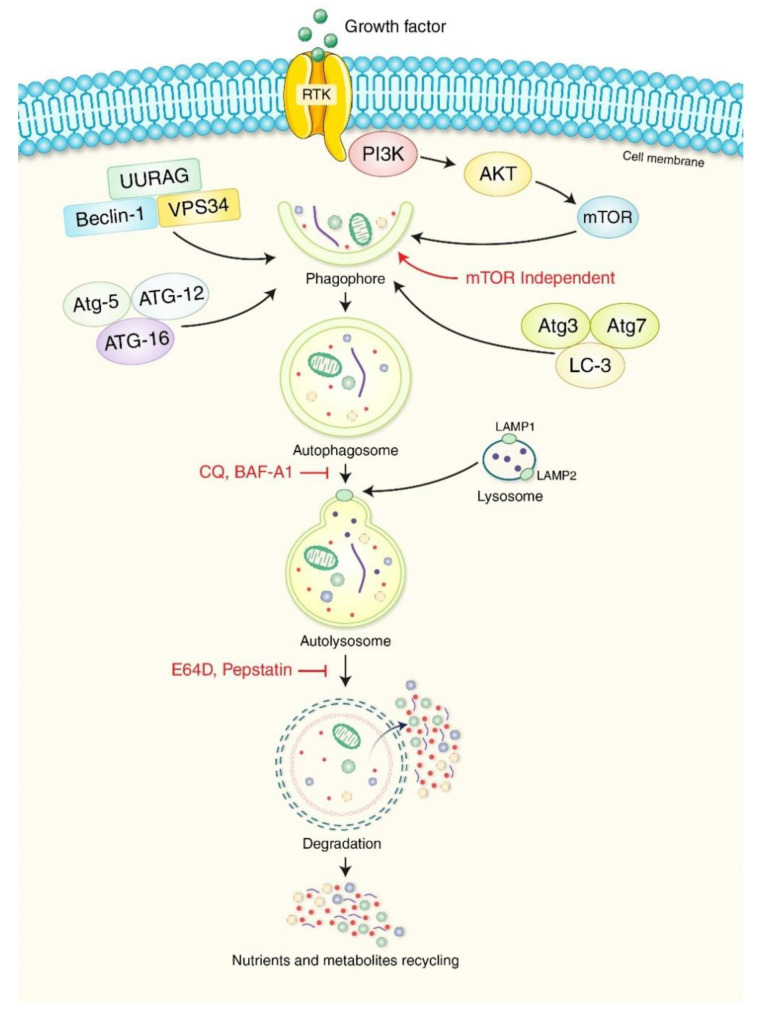
Molecular mechanism of autophagy. The autophagy process begins through the formation of phagophore structures. PI3K, protein kinase B (AKT), and mTOR are important for phagophore initiation. The BECN1, UVRAG, and VPS34 complex facilitate phagophore formation. Phagophore nucleation leads to autophagosome formation. Binding between mature autophagosomes and lysosomes leads to autolysosome formation. Chloroquine (CQ), Bafilomycin A1 (BAF-A1), E64D, and Pepstatin inhibit the binding of lysosomes and autophagosomes. Eventually, autolysosomes are eliminated by acid hydrolases, producing nutrients and recycling metabolites.

**Figure 2 biomedicines-10-01459-f002:**
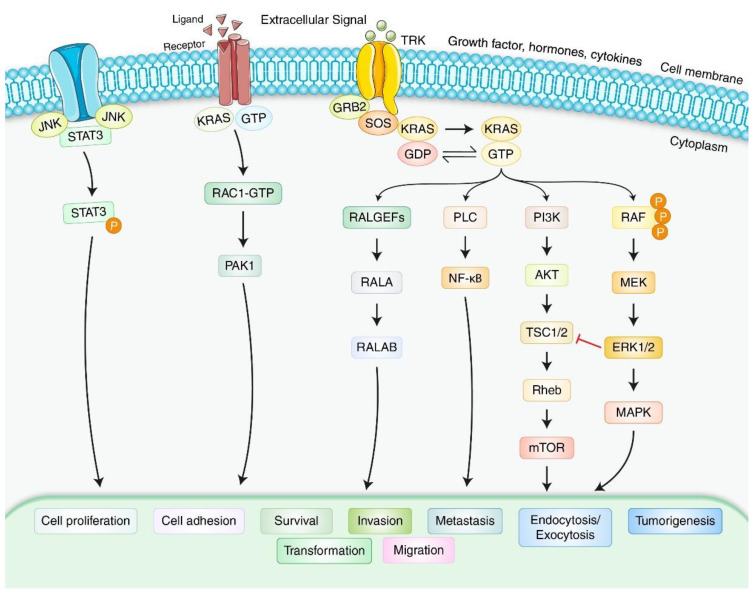
Formation of PC via several signaling pathways. Receptor tyrosine kinases (TRK) employ KRAS guanine nucleotide exchange factors (GEFs) such as son-of-sevenless (SOS) to activate GEFs and GTPase activating proteins (GAPs) that convert KRAS between its GTP- and GDP-bound states. The constitutive GDP-bound state triggers numerous downstream molecules in PDAC, leading to proliferation, invasion, metastasis, tumorigenesis, and migration. Ligand-mediated receptor KRAS-RAC1-GTP and JNK-STAT3 signaling are involved in activating PDAC.

**Figure 3 biomedicines-10-01459-f003:**
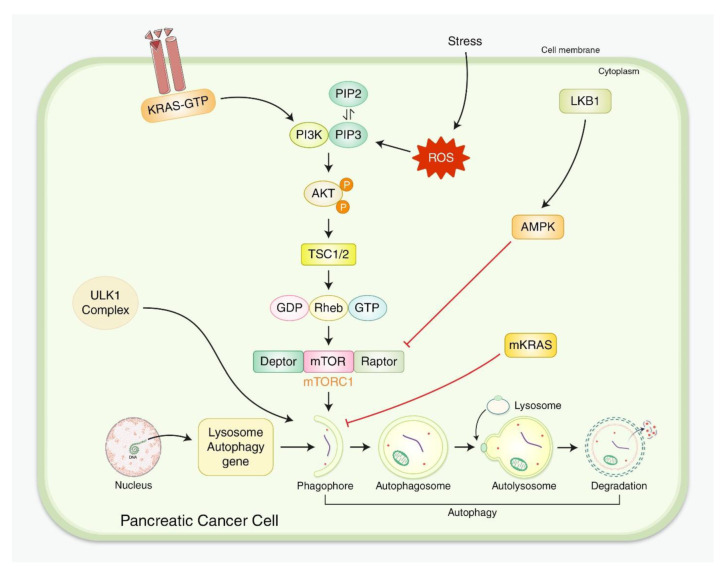
The molecular pathways involved in regulating PC metabolism through autophagy. KRAS stimulation includes a GEF-triggered GDP–GTP exchange, permitting KRAS-GDP phosphorylation to GTP, leading to effector functions, including activation of PI3K-AKT-mTOR-targeted autophagy activation. The LKB1–AMPK pathway inhibits mTORC1, causing autophagy initiation in PC cells. Moreover, several nuclear autophagy-related genes initiate phagophore formation, activating autophagy in PC cells.

**Figure 4 biomedicines-10-01459-f004:**
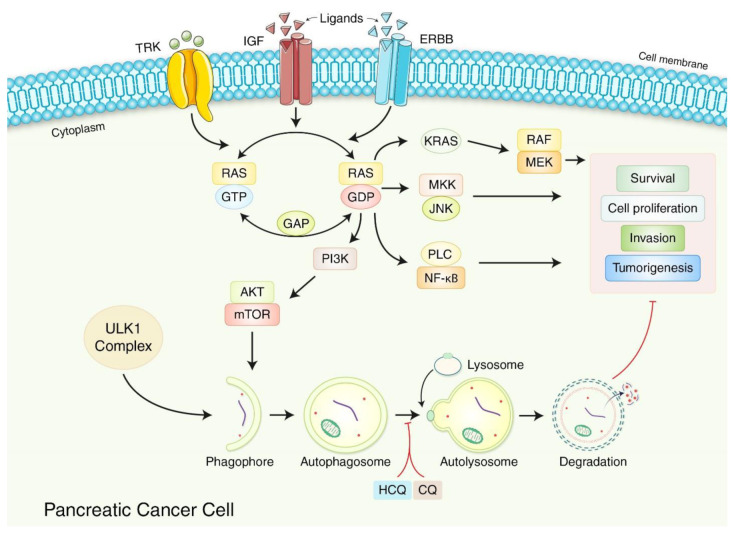
Autophagy-related PDAC therapies associated with the neuromere signaling pathway.

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
