# Peer review of "Therapeutic Aspects and Molecular Targets of Autophagy to Control Pancreatic Cancer Management"

_biomedicines, 2022, doi:10.3390/biomedicines10061459_

Round 1
Reviewer 1 Report
In this review, the evidences have been implicated for the tumorigenic and protective role of autophagy in PC tumorigenesis were summarized. A description of recently advances understanding of how autophagy might be regulated and control in PDAC was also briefly discussed. This paper is very well written but it needs some input of the authors in terms of opinion (challenges and opportunities). On the other hand, this paper is lacking some information about new drugs recently approved (section 6, see EMA or FDA website) and they should be appointed as a potential solution. Conclusions should be better supported specially when authors stated that "developing a straightforward, durable, and dependable framework for measuring autophagy in using human samples (blood and tumor) would be a crucial step forward in constructing clinical trials which evaluate autophagy attenuation and patient identification using biomarkers." How it can be made?
Author Response
In this review, the evidences have been implicated for the tumorigenic and protective role of autophagy in PC tumorigenesis were summarized. A description of recently advances understanding of how autophagy might be regulated and control in PDAC was also briefly discussed. This paper is very well written but it needs some input of the authors in terms of opinion (challenges and opportunities). On the other hand, this paper is lacking some information about new drugs recently approved (section 6, see EMA or FDA website) and they should be appointed as a potential solution. Conclusions should be better supported specially when authors stated that "developing a straightforward, durable, and dependable framework for measuring autophagy in using human samples (blood and tumor) would be a crucial step forward in constructing clinical trials which evaluate autophagy attenuation and patient identification using biomarkers." How it can be made?
>> (Response) First of all, we would like to express our sincere gratitude for the time and effort the reviewer had put into reviewing our manuscript. We have incorporated changes based on the reviewer comments provided in the manuscript which revised parts are highlighted by BLUE color in the entire revised manuscript.
Regarding challenges and opportunities with new drugs recently approved, we added recent information in section 6 from EMA and FDA which has been appointed as a potential solution of PDAC (page 15 & 16, line 371-387).
In conclusion, we modified the sentence with better information in overall conclusion part and adding with future direction as suggested by reviewer 2 (page 20 & 21).
Reviewer 2 Report
The authors described the evidences have been implicated for the tumorigenic and protective role of autophagy in PC tumorigenesis was summarized which followed by a description of recently advances understanding of how autophagy might be regulated and control in PDAC. I think this review is timely and interesting, and could be published after the following minor modification.
- For autophagy, the literature about autophagy-controlled ATP might be useful.Adv. Funct. Mater.2022, DOI: 10.1002/adfm.202200801。
- What is the full name for ROS in page 2?
- For the conclusion, is it possible that the authors give the detailed future direction for this important research field?
Author Response
The authors described the evidences have been implicated for the tumorigenic and protective role of autophagy in PC tumorigenesis was summarized which followed by a description of recently advances understanding of how autophagy might be regulated and control in PDAC. I think this review is timely and interesting, and could be published after the following minor modification.
>> (Response) First of all, we would like to express our sincere gratitude for the time and effort the reviewer had put into reviewing our manuscript. We have incorporated changes based on the reviewer comments provided in the manuscript which revised parts are highlighted by BLUE color in the entire revised manuscript.
- For autophagy, the literature about autophagy-controlled ATP might be useful. Adv. Funct. Mater.2022, DOI: 10.1002/adfm.202200801。
>> (Response) We added role of ATP to control autophagy in the section ‘3. Molecular pathway of autophagy process’ with some new and mentioned reference (page 5, line 122-134).
2. What is the full name for ROS in page 2?
>> (Response) ROS (Reactive oxygen species) mentioned in page 4, line 81.
3. For the conclusion, is it possible that the authors give the detailed future direction for this important research field?
>> (Response) we modified and changed all conclusion part and added one more paragraph future direction in the last portion of conclusion (page 21, line 517-527).